# ACTIONABLE RECOURSE GUIDED BY USER PREFERENCE

## ABSTRACT

The growing popularity of machine learning models has led to their increased application in domains directly impacting human lives. In critical fields such as healthcare, banking, and criminal justice, tools that ensure trust and transparency are vital for the responsible adoption of these models. One such tool is *actionable recourse* (AR) for negatively impacted users. AR describes recommendations of cost-efficient changes to a user's *actionable* features to help them obtain favorable outcomes. Existing approaches for providing recourse optimize for properties such as proximity, sparsity, validity, and distance-based costs. However, an often-overlooked but crucial requirement for actionability is a consideration of *User Preference* to guide the recourse generation process. Moreover, existing works considering a user's preferences require users to precisely specify their costs for taking actions. This requirement raises questions about the practicality of the corresponding solutions due to the high cognitive loads imposed. In this work, we attempt to capture user preferences via soft constraints in three simple forms: *i) scoring continuous features, ii) bounding feature values* and *iii) ranking categorical features*. We propose an optimization framework that is sensitive to user preference and a gradient-based approach to identify *User Preferred Actionable Recourse (UP-AR)*. We empirically demonstrate the proposed approach's superiority in adhering to user preference while maintaining competitive performance in traditional metrics with extensive experiments.

## 1 INTRODUCTION

*Actionable Recourse (AR)* (Ustun et al., 2019) is the ability of an individual to obtain the desired outcome from a fixed Machine Learning (ML) model. Several domains such as lending (Siddiqi, 2012), insurance (Scism, 2019), resource allocation (Chouldechova et al., 2018; Shroff, 2017) and hiring decisions (Ajunwa et al., 2016) are required to suggest recourses to ensure the trust of the decision system in place; in such scenarios, it is critical to ensure actionability in recourse (otherwise the suggestions are pointless). Consider an individual named Alice who applies for a loan, and the bank, which uses an ML-based classifier, denies it. Naturally, Alice asks - *What can I do to get the loan?* The inherent question is what action she must take to obtain the loan in the future. *Counterfactual explanation* introduced in *Wachter* (Wachter et al., 2017) provides a *what-if* scenario to alter the model's decision. AR further aims to provide Alice with a *feasible* action. A feasible action is both actionable by Alice (meaning she can reasonably execute the directed plan) and suggests as low-cost modifications as possible.

While some features (such as age or sex) are inherently inactionable, Alice's personalized constraints may also limit her ability to take action on the suggested recourse (such as a strong reluctance to secure a co-applicant). We call these localized constraints *User Preferences*, synonymous to user-level constraints introduced as *local feasibility* by Mahajan et al. (2019). Figure 1 illustrates the motivation behind UP-AR. Notice how similar individuals can prefer contrasting recourse.

*Actionability*, as we consider it, is centered explicitly around individual preferences, and similar recourses provided to two individuals (Alice and Bob) with identical feature vectors may not necessarily be equally actionable. Most existing methods of finding actionable recourse are restricted to *omission* of features from the *actionable feature set* which Alice does not prefer to act upon, and *box constraints* (Mothilal et al., 2020) in the form of bounds on feature actions.

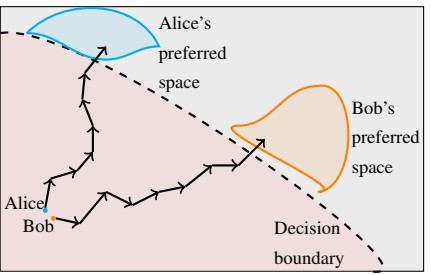

Figure 1: Illustration of UP-AR. Similar individuals Alice and Bob with contrasting preferences can have different regions of desired feature space for a recourse.

Table 1: A hypothetical actionable feature set of adversely affected individuals sharing similar features and corresponding suggested actions by AR and UP-AR. UP-AR provides personalized recourses based on individual user preferences.

| Actionable Features | Curr. val. | UP–AR values | |
|---|---|---|---|
| | | Alice | Bob |
| LoanDuration | 18 | 8 | 17 |
| LoanAmount | $1940 | $1840 | $1200 |
| HasGuarantor | 0 | 0 | 1 |
| HasCoapplicant | 0 | 1 | 0 |

In this study, we discuss three forms of user preferences and propose a formulation capturing such idiosyncrasies. The proposed score-based preference mechanism can easily communicate with an individual user, providing a seamless feasible recourse. We argue that communicating in terms of preference scores improves the *explainability of a recourse* generation mechanism, which ultimately *improves trust* in an ML model. We provide a hypothetical example of UP-AR's ability to adapt to individual preferences in Table 1. However, there is a lack of contemporary research in understanding Alice's individual preference, which may be contrary to Bob's preference. It is also worth mentioning that the existing method's rigid modifications on actionable features limit the possibility of identifying a recourse due to diminished actionable space.

Motivated by the above limitation, we intend to capture soft user preferences along with hard constraints and identify recourse based on local desires without affecting the success rate of identifying recourse. For example, consider Alice prefers to have $80\%$ of fractional cost from loan duration and only $20\%$ from the loan amount, meaning she prefers to have recourse with a minor reduction in the loan amount. Such recourse enables Alice to get the benefits of a loan on her terms. We study the problem of providing *user preferred recourse* by solving a custom optimization for individual user-based preferences. Our contributions are consolidated as follows:

- We start by enabling Alice to provide three types of user preferences: i) *Scoring*, ii) *Ranking*, and iii) *Bounding*. We embed them into an optimization function to guide the recourse generation mechanism.

- We then present our approach called *User Preferred Actionable Recourse (UP-AR)* to identify a recourse instead of overwhelming her with a variety of recourse options. Our approach highlights a cost correction step to address the *redundancy* induced by our method.

- We also consolidate performance metrics with empirical results of UP-AR across multiple datasets and compare them with state-of-art techniques.

## 1.1 RELATED WORKS

Several methods exist to identify counterfactuals, such as FACE (Poyiadzi et al., 2020), which uses the shortest path to identify counterfactual explanations from high-density regions, and Growing Spheres (GS) (Laugel et al., 2017) which employs random sampling within increasing hyperspheres for finding counterfactuals. CLUE (Antoran et al., 2021) identifies counterfactuals with low uncertainty in terms of the classifier's entropy within the data distribution. Similarly, manifold-based CCHVAE (Pawelczyk et al., 2020) generates high-density counterfactuals through the use of a latent space model. However, there is often no guarantee that the *what-if* scenarios identified by these methods are attainable.

AR observes that a feasible recourse achieves good results in all the performance metrics. Existing research focuses on providing feasible recourses, yet comprehensive literature on understanding and incorporating user preferences within the recourse generation mechanism is lacking. It is worth mentioning that instead of better understanding user preferences, Mothilal et al. (2020) provides a user with diverse recourse options and hopes that the user will benefit from at least one. The importance of diverse recourse recommendations have already been explored in recent works (Wachter et al.,

2017; Mothilal et al., 2020; Russell, 2019), which can be summarized as increasing the chances of actionability as intuitively observed in the domain of unknown user preferences (Karimi et al., 2021). Karimi et al. (2020) and Cheng et al. (2020) resolves uncertainty in user cost function by inducing *diversity* in the suggested recourses. Interestingly, only 16 out of the 60 recourse methods explored in the survey by Karimi et al. (2021) include diversity as a constraint where diversity is measured in terms of distance metrics. Alternatively, studies like Ustun et al. (2019); Rawal & Lakkaraju (2020); Cui et al. (2015) optimize on a universal cost function. This assumption does not capture individual idiosyncrasies and preferences crucial for actionability.

Initial efforts of eliciting user preferences include recent work by De Toni et al. (2022). The authors provide an interactive human-in-the-loop approach, where a user continuously interacts with the system. However, learning user preferences by asking them to select from one of the *partial interventions* (De Toni et al., 2022) provided is a derivative of providing a diverse set of recourse candidates. In this work, we consider fractional cost as a means to communicate with Alice instead of recourse generation procedure or the nature of the cost function. *The fractional cost of a feature refers to the fraction of cost incurred from a feature i out of the total cost of the required intervention.*

The notion of user preference or user-level constraints was previously studied as *local feasibility* (Mahajan et al., 2019). Since users can not precisely quantify the cost function (Rawal & Lakkaraju, 2020), Yadav et al. (2021) diverged from the assumption of a universal cost function and optimizes over the distribution of cost functions. We argue that the inherent problem of feasibility can be solved by capturing and understanding Alice's recourse preference and adhering to her constraints which can vary between *Hard Rules* such as unable to bring a co-applicant and *Soft Rules* such as hesitation to reduce the amount, which should not be interpreted as unwillingness. This study aims to capture individual constraints and provide a user-preferred recourse. To the best of our knowledge, this is the first study to capture individual idiosyncrasies in the recourse generation optimization problem to improve feasibility.

## 2 PROBLEM FORMULATION

Consider a binary classification problem where each instance represents an individual's feature vector $\mathbf{x} = [\mathbf{x}_1, \mathbf{x}_2, \ldots, \mathbf{x}_D]$ and an associated binary label $\mathbf{y} \in \{-1, +1\}$. We are given a model $f(\mathbf{x})$ to classify $\mathbf{x}$ into either $-1$ or $+1$. Let $f(\mathbf{x}) = +1$ be the desirable output of $f(\mathbf{x})$ for Alice. However, Alice was assigned an undesirable label of $-1$ by $f$. We consider the problem of suggesting an action $\mathbf{r} = [\mathbf{r}_1, \mathbf{r}_2, \ldots, \mathbf{r}_D]$ such that $f(\mathbf{x} + \mathbf{r}) = +1$. Since suggested recourse only requires actions to be taken on *actionable features* denoted by $F_A$, we have $\mathbf{r}_i \equiv 0 : \forall i \notin F_A$. We further split $F_A$ into *continuous actionable features* $F_{con}$ and *categorical actionable features* $F_{cat}$ based on the feature domain. A low-cost action $\mathbf{r}$ is obtained by solving the following optimization, where $userCost(\mathbf{r}, \mathbf{x})$ is any predefined cost function of taking an action $\mathbf{r}$ specified by Alice.

$$\min_{\mathbf{r}} \; userCost(\mathbf{r}, \mathbf{x}) \quad s.t. \; userCost(\mathbf{r}, \mathbf{x}) = \sum_{i \in F_A} userCost(\mathbf{r}_i, \mathbf{x}_i) \; \text{and} \; f(\mathbf{x} + \mathbf{r}) = +1. \quad (1)$$

### 2.1 CAPTURING INDIVIDUAL IDIOSYNCRASIES

A crucial step for generating recourse is identifying *local feasibility* constraints captured in terms of individual user preferences. In this study, we assume that every user provides their preferences in three forms. Every continuous actionable feature $i \in F_{con}$ is associated with a *preference score* $\Gamma_i$ obtained from the affected individual. Additional preferences in the form of feature value bounds and ranking for preferential treatment of categorical features are also requested from Alice.

**User Preference Type I (Scoring continuous features):** User preference for continuous features are captured in $\Gamma_i \in [0, 1] : \forall i \in F_{con}$ subject to $\sum_{i \in F_{con}} \Gamma_i = 1$. Such *soft constraints* capture the user's preference without omitting the feature from the actionable feature set. $\Gamma_i$ refers to the fractional cost of action Alice prefers to incur from a continuous feature $i$. For example, consider $F_{con} = \{LoanDuration, LoanAmount\}$ with corresponding user-provided scores $\Gamma = \{0.8, 0.2\}$ implying that Alice prefers to incur $80\%$ of fractional feature cost from taking action on *LoanDuration*, while only $20\%$ of fractional cost from taking action on *LoanAmount*. Here, Alice prefers reducing *LoanDuration* to *LoanAmount* and providing recourse in accordance improves actionability.

Figure 2: Framework of UP-AR. Successful recourse candidates; $\mathbf{r}^{(\cdot)}$, $\bar{\mathbf{r}}^{(\cdot)}$ are colored in pink.

**User Preference Type II (Bounding feature values):** Users can also provide constraints on values for individual features in $F_A$. These constraints are in the form of lower and upper bounds for individual feature values represented by $\underline{\delta_i}$ and $\overline{\delta_i}$ for any feature $i$ respectively. These constraints are used to discretize the steps. For a continuous feature $i$, action steps can be discretized into pre-specified step sizes of $\Delta_i = \{s : s \in [\underline{\delta_i}, \overline{\delta_i}]\}$. For categorical features, steps are defined as the feasible values a feature can take. For all categorical features we define, $\Delta_i = \{\underline{\delta_i}, \ldots, \overline{\delta_i}\} : \forall i \in F_{cat}$ representing the possible values for categorical feature $i$.

**User Preference Type III (Ranking categorical features):** Users are also asked to provide a ranking function $\mathcal{R} : F_{cat} \rightarrow \mathbb{Z}^{+1}$ on $F_{cat}$. Let $\mathcal{R}_i$ refers to the corresponding rank for a categorical feature $i$. Our framework identifies recourse by updating the candidate action based on the ranking provided. For example, consider $F_{cat} = \{HasCoapplicant, HasGuarantor, CriticalAccountOrLoansElsewhere\}$ for which Alice ranks them by $\{3, 2, 1\}$. The recourse generation system considers suggesting an action on *HasGuarantor* before *HasCoapplicant*. Ranking preferences can be easily guaranteed by a simple override in case of discrepancies while finding a recourse.

## 2.2 PROPOSED OPTIMIZATION

We depart from capturing a user's cost of feature action and instead obtain their preferences for each feature. We elicit three forms of preferences detailed in the previous section and iteratively take steps in the action space. We propose the following optimization over the basic predefined steps based on the *user preferences*. Let us denote the inherent hardness of feature action $\mathbf{r}_i$ for feature value $\mathbf{x}_i$ using $cost(\mathbf{r}, \mathbf{x})$ which can be any cost function easily communicable to Alice. Here, $cost\left(\mathbf{r}_i^{(t)}, \mathbf{x}_i\right)$ refers to a "universal" cost of taking an action $\mathbf{r}_i^{(t)}$ for feature value $\mathbf{x}_i$ at step $t$. Note that this cost function or quantity differs from the *userCost* $(\cdot, \cdot)$ function specified earlier. This quantity is capturing the inherent difficulty of taking an action.

$$\max_{\mathbf{r}} \quad \sum_{i \in F_A} \frac{\Gamma_i}{cost(\mathbf{r}_i, \mathbf{x}_i)} \qquad \text{(Type I)}$$

$$s.t. \quad f(\mathbf{x} + \mathbf{r}) = +1$$
$$\Gamma_i = 0 : \forall i \notin F_A \qquad \text{(actionability)}$$
$$\Gamma_j = 1 : \forall j \in F_{cat}$$
$$\mathbf{r}_i \in \Delta_i : i \in F_A \qquad \text{(Type II)}$$
$$\mathbf{1}\{\mathbf{r}_i > 0\} \geq \mathbf{1}\{\mathbf{r}_j > 0\} : \mathcal{R}_i \geq \mathcal{R}_j \; \forall i, j \in F_{cat} \qquad \text{(Type III)}$$

The proposed method minimizes the cost of a recourse weighted by $\Gamma_i$ for all actionable features. We discuss the details of our considerations of cost function in Section 3.1. The order preference of categorical feature actions can be constrained by restrictions while finding a recourse. The next section introduces UP-AR as a stochastic solution to the proposed optimization.

## 3 USER PREFERRED ACTIONABLE RECOURSE (UP-AR)

Our proposed solution, User Preferred Actionable Recourse (UP-AR), consists of two stages. The first stage generates a candidate recourse by following a connected gradient-based iterative approach. The second stage then improves upon the *redundancy* metric of the generated recourse for better actionability. The details of UP-AR are consolidated in Algorithm 1 and visualized in Figure 2.

### 3.1 STAGE 1: STOCHASTIC GRADIENT-BASED APPROACH

Poyiadzi et al. (2020) identifies a counterfactual by following a high-density connected path from the feature vector $\mathbf{x}$. With a similar idea, we follow a connected path guided by the user's preference to identify a feasible recourse. We propose incrementally updating the candidate action with a predefined step size to solve the optimization. At each step $t$, a candidate intervention is generated, where any feature $i$ is updated based on a Bernoulli trial with probability $I_i^{(t)}$ derived from user preference scores and the cost of taking a predefined step $\delta_i^{(t)}$ using the following procedure:

$$I_i^{(t)} \sim Bernoulli\left(\sigma\left(z_i^{(t)}\right)\right) \quad \text{where} \quad \sigma\left(z_i^{(t)}\right) = \frac{e^{z_i^{(t)}/\tau}}{\sum_{j \in F_A} e^{z^{(t)}/\tau}}, \quad z_i^{(t)} = \frac{\Gamma_i}{cost\left(\mathbf{r}_i^{(t)}, \mathbf{x}_i\right)} \quad (2)$$

With precomputed costs for each step, *weighted inverse cost* is computed for each feature, and these values are mapped to a probability distribution using a function like softmax. *Softmax* gives a probabilistic interpretation $P\left(I_i^{(t)} = 1 | z_i^{(t)}\right) = \sigma\left(z_i^{(t)}\right)$ by converting $z_i^{(t)}$ scores into probabilities.

We leverage the idea of *log percentile shift* from AR to determine the cost of action since it is easier to communicate with the users in terms of percentile shifts. Specifically, we follow the idea and formulation in Ustun et al. (2019) to define the cost:

$$cost\left(\mathbf{r}_i, \mathbf{x}_i\right) = log\left(\frac{1 - Q_i\left(\mathbf{x}_i + \mathbf{r}_i\right)}{1 - Q_i\left(\mathbf{x}_i\right)}\right) \quad (3)$$

were $Q_i\left(\mathbf{x}_i\right)$ representing the *percentile* of feature $i$ with value $\mathbf{x}_i$ is a score below which $Q_i\left(\mathbf{x}_i\right)$ percentage of scores fall in the frequency distribution of feature values in the target population.

We adapt and extend the idea that counterfactual explanations and adversarial examples (Szegedy et al., 2014) have a similar goal but with contrasting intention (Pawelczyk et al., 2022). A popular approach to generating adversarial examples (Goodfellow et al., 2014) is by using a gradient-based method. We employ the learning of adversarial example generation to determine the direction of feature modification in UP-AR. Jacobian matrix is used to measure the local sensitivity of outputs with respect to each input feature. Consider that $f : \mathbb{R}^D \to \mathbb{R}^K$ maps a $D$-dimensional feature vector to a $K$-dimensional vector, such that each of the partial derivatives exists. For a given $\mathbf{x} = [\mathbf{x}_1, \ldots, \mathbf{x}_i, \ldots, \mathbf{x}_D]$ and $f(\mathbf{x}) = [f_{[1]}(\mathbf{x}), \ldots, f_{[j]}(\mathbf{x}), \ldots, f_{[K]}(\mathbf{x})]$, the Jacobian matrix of $f$ is defined to be a $D \times K$ matrix denoted by $\mathbf{J}$, where each $(j, i)$ entry is $\mathbf{J}_{j,i} = \frac{\partial f_{[j]}(\mathbf{x})}{\partial \mathbf{x}_i}$. For a neural network (NN) with at least one hidden layer, $\mathbf{J}_{j,i}$ is obtained using the chain rule during backpropagation. For an NN with one hidden layer represented by *weights* $\{w\}$, we have:

$$\mathbf{J}_{j,i} = \frac{\partial f_{[j]}(\mathbf{x})}{\partial \mathbf{x}_i} = \sum_l \frac{\partial f_{[l]}(\mathbf{x})}{\partial a_l} \frac{\partial a_l}{\partial \mathbf{x}_i} \quad \text{where} \quad a_l = \sum_i w_{li} \mathbf{x}_i \quad (4)$$

Where in Equation 4, $a_l$ is the output (with possible activation) of the hidden layer and $w_l$ is the weight of the node $l$. Notice line 4 in Algorithm 1 which *updates the candidate* action for a feature $i$ at step $t$ as:

$$\mathbf{r}_i^{(t)} = \mathbf{r}_i^{(t-1)} + \text{sign}\left(\mathbf{J}_{+1,i}^{(t)}\right) \circ I_i^{(t)} \circ \delta_i^{(t)} \quad (5)$$

Here $\mathbf{m} \circ \mathbf{n}$ is the Hadamard product of any two vectors $\mathbf{m}$ and $\mathbf{n}$. Following the traditional notation of a binary classification problem and with a bit of abuse of notation $-1 \to 1, +1 \to +1$, $\text{sign}\left(\mathbf{J}_{+1,i}^{(t)}\right)$ captures the direction of the feature change at step $t$. This direction is iteratively calculated, and additional constraints such as non-increasing or non-decreasing features can be placed at this stage.

### 3.1.1 CALIBRATING FREQUENCY OF CATEGORICAL ACTIONS

We employ *temperature scaling* (Guo et al., 2017) parameter $\tau$ observed in Equation 2 to calibrate UP-AR's recourse generation cost. Categorical actions with fixed step sizes are expensive, especially for binary categorical values. Hence, tuning the frequency of categorical suggestions can significantly impact the overall cost of a recourse. $\tau$ controls the frequency with which categorical actions are suggested in UP-AR.

---

**Algorithm 1** User Preferred Actionable Recourse (UP-AR)

---

**Input**: Model $f$, user feature vector $\mathbf{x}$, cost function $cost\left(\cdot,\cdot\right)$, step size $\Delta_i : \forall i \in F_A$, maximum steps $T$, action $\mathbf{r}$ initialized to $\mathbf{r}^{(0)}$, fixed $\tau$, $t = 1$.

1: **while** $t \leq T$ or $f\left(\mathbf{x} + \mathbf{r}^{(t)}\right) \neq +1$ **do**

2:      $z_i^{(t)} = \frac{\Gamma_i}{cost\left(\mathbf{r}_i^{(t)}, \mathbf{x}_i\right)}$   $\forall i$

3:      $I_i^{(t)} \sim Bern(\sigma(z_i^{(t)}))$ : where $\sigma(z_i^{(t)}) = \frac{e^{z_i^{(t)}/\tau}}{\sum_{j \in F_A} e^{z^{(t)}/\tau}}$

4:      $\mathbf{r}_i^{(t)} = \mathbf{r}_i^{(t-1)} + \text{sign}\left(\mathbf{J}_{+1,i}^{(t)}\right) \circ I_i^{(t)} \circ \delta_i^{(t)} : \forall i \in F_A$

5:      $t = t + 1$

6: Let $\hat{t}$ be the smallest step such that $f(\mathbf{x} + \mathbf{r}^{(\hat{t})}) = +1$ and initialize $t = \hat{t}$

7: **if** $\exists i \in F_{cat} : \mathbf{r}_i^{(t)} > 0$ **then**

8:      **while** $f(\mathbf{x} + \bar{\mathbf{r}}^{(t)} = +1)$ **do**

9:         $\bar{\mathbf{r}}^{(t)} = \mathbf{r}^{(t)}$;    $\bar{\mathbf{r}}_i^{(t)} = \mathbf{r}_i^{(\hat{t})}$ $\forall i \in F_{cat}$;    $t = t - 1$

10: **return** $\bar{\mathbf{r}}^{(t)}$ as action $\mathbf{r}$

---

To study the effect of $\tau$ on overall cost, we train a Logistic Regression (LR) model on a processed version of *German* (Bache & Lichman, 2013) dataset and generate recourses for the 155 individuals who were denied credit. Figure 10 in Appendix C.3 depicts that the cost gradually decreases with decreasing $\tau$ since the marginal probability of suggesting a categorical feature change is diminished. Hence, without affecting the success rate of recourse generation, the overall cost of generating recourses can be brought down by decreasing $\tau$. In simple terms, with a higher $\tau$, UP-AR frequently suggests recourses with expensive categorical actions. After the strategic generation of an intervention, we move to the next step of cost correction to improve upon the redundancy metric.

## 3.2 STAGE 2: REDUNDANCY AND COST CORRECTION (CC)

In our experiments, we observe that once an expensive action is recommended for a categorical feature, some of the previously generated action steps might become redundant. Consider an LR model trained on the processed *german* dataset. Let $F_A = \{LoanDuration, LoanAmount, HasGuarantor\}$ out of all the 26 features, where *HasGuarantor* is a binary feature which represents the user's ability to get a guarantor for the loan. Stage 1 takes several steps over *LoanAmount* and *LoanDuration* before recommending to update *HasGuarantor*. These steps are based on the feature action probability from Equation 2. Since categorical feature updates are expensive and occur with relatively low probability, Stage 1 finds a low-cost recourse by suggesting low-cost steps more frequently in comparison with high-cost steps.

Table 2: Redundancy corrected recourse for a hypothetical individual.

| Features to change | Current values | Stage 1 values | Stage 2 values |
|---|---|---|---|
| LoanDuration | 18 | 8 | 12 |
| LoanAmount | $1940 | $1040 | $1540 |
| HasGuarantor | 0 | 1 | 1 |

Once a high-cost feature such as *HasGuarantor* is recommended to update, some of the previous low-cost steps may be redundant, which can be rectified by tracing back previous steps. Notice that the cost correction (CC) procedure is only triggered when UP-AR suggests a categorical action. Consider a scenario such that $\exists i \in F_{cat} : \mathbf{r}_i^{(T)} > 0$ for a recourse obtained after $T$ steps in Stage 1. The CC procedure updates all the intermediary recourse candidates to reflect the categorical changes i.e., $\forall i \in F_{cat} : \mathbf{r}_i^{(T)} > 0$, we update $\mathbf{r}_i^{(t)} = \mathbf{r}_i^{(T)} : \forall t \in \{1, 2, \ldots, T-1\}$ to obtain $\bar{\mathbf{r}}^{(t)}$. We then perform a linear retracing procedure to return $\bar{\mathbf{r}}^{(t)}$ such that $f\left(\mathbf{x} + \bar{\mathbf{r}}^{(t)}\right) = +1$ for the smallest $t$.

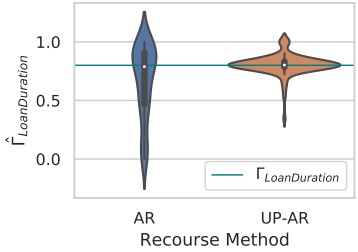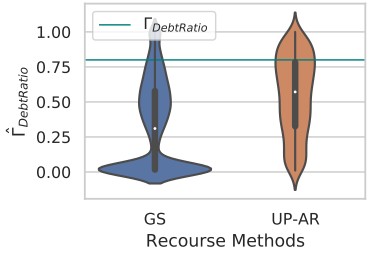

Figure 3: AR and UP-AR's distribution of $\hat{\Gamma}_{LoanDuration}$ for a *Logistic Regression* model trained on *German*.

Figure 4: GS and UP-AR's distribution of $\hat{\Gamma}_{DebtRatio}$ for a *Neural Network* model trained on *GMSC*.

## 4 DISCUSSION AND ANALYSIS

In this section, we analyze the user preference performance of UP-AR. For simplicity, a user understands cost in terms of log percentile shift from her initial feature vector described in Section 3. Let $\hat{\Gamma}_i$ be the observed fractional cost for feature $i$ formally defined in Equation 6. Any cost function can be plugged into UP-AR with no restrictions. A user prefers to have $\Gamma_i$ fraction of the total desired percentile shift from feature $i$. Consider $F_A = \{LoanDuration, LoanAmount\}$ and let the corresponding user scores provided by all the adversely affected individuals be: $\Gamma = \{0.8, 0.2\}$. Here, "Denied loan applicants prefers reducing *LoanDuration* to *LoanAmount* by $8:2$." Figure 3 shows the frequency plot of feature cost ratio for feature *LoanDuration* out of total incurred cost from *LoanDuration* and *LoanAmount*. i.e., $y-$axis represents $\hat{\Gamma}_i$. Also, Figure 4 further shows the fractional cost of feature *DebtRatio* for recourses obtained for a NN based model trained on *Give Me Some Credit (GMSC)* dataset. **These experiments signify the adaptability of UP-AR to user preferences and provides evidence that UP-AR's distribution of $\hat{\Gamma}_i$ is centered around $\Gamma_i$.**

**Lemma 1 (Cost guarantee)** *Consider UP-AR identified recourse $\mathbf{r}$ for an individual $\boldsymbol{x}$. If $C_{i,min}^{(T^*)}$ and $C_{i,max}^{(T^*)}$ represent the minimum and maximum cost of any step for feature $i$ until $T^*$, then:*

$$\mathbb{E}\left[cost\left(\mathbf{r}_i, \boldsymbol{x}_i\right)\right] \leq T^* \sigma \left(\frac{\Gamma_i}{C_{i,min}^{(T^*)}}\right) C_{i,max}^{(T^*)}$$

Lemma 1 implies that the expected cost $\mathbb{E}\left[cost\left(\mathbf{r}_i, \mathbf{x}_i\right)\right]$, specifically for a continuous feature action is positively correlated to the probabilistic interpretation of user preference scores. Hence $\mathbf{r}$ satisfies users critical Type I constraints in expectation. Recall that Type II and III constraints are also applied at each step $t$. Lemma 1 signifies that UP-AR adheres to user preferences and thereby increases the actionability of a suggested recourse.

**Corollary 1** *For UP-AR with a linear $\sigma\left(\cdot\right)$, predefined steps with equal costs and $cost\left(\mathbf{r}, \boldsymbol{x}\right) = \sum_{i \in F_A} cost\left(\mathbf{r}_i, \boldsymbol{x}_i\right)$, total expected cost after $T^*$ steps is : $\mathbb{E}\left[cost\left(\mathbf{r}, \boldsymbol{x}\right)\right] = T^* \sum_{i \in F_A} \sigma\left(\Gamma_i\right).$*

Corollary 1 states that with strategic selection of $\sigma\left(\cdot\right)$, $\delta_{\cdot}^{(\cdot)}$ and $cost\left(\cdot, \cdot\right)$, UP-AR can also tune the total cost of suggested actions. In the next section, we will compare multiple recourses based on individual user preferences for a randomly selected adversely affected individual.

### 4.1 CASE STUDY OF INDIVIDUALS WITH SIMILAR FEATURES BUT DISPARATE PREFERENCES

Given an LR model trained on *german* dataset and Alice, Bob and Chris be three adversely affected individuals. $F_A = \{LoanDuration, LoanAmount, HasGuarantor\}$ and corresponding user preferences are provided by the users. In Table 3, we consolidate the corresponding recourses generated for the specified disparate sets of preferences. From Table 3 we emphasize the ability of UP-AR to generate a variety of user-preferred recourses based on their preferences, whereas AR always provides the same low-cost recourse for all the individuals. The customizability of feature actions for individual users can be found in the table. When the Type I score for *LoanAmount* is $0.8$, UP-AR prefers decreasing loan amount to loan duration. Hence, the loan amount is much lesser for Chris than for Alice and Bob.

Table 3: Recourses generated by UP-AR for similar individuals with a variety of preferences.

| Features to change | Current values | AR values | Alice | | Bob | | Chris | |
|---|---|---|---|---|---|---|---|---|
| | | | User Pref. | UP-AR values | User Pref. | UP-AR values | User Pref. | UP-AR values |
| LoanDuration | 30 | 25 | 0.8 | 20 | 0.8 | 10 | 0.2 | 27 |
| LoanAmount | $8072 | $5669 | 0.2 | $7372 | 0.2 | $6472 | 0.8 | $5272 |
| HasGuarantor | 0 | 1 | 1 | 1 | 0 | 0 | 1 | 1 |

## 5 EMPIRICAL EVALUATION

In this section, we demonstrate empirically: 1) that UP-AR respects $\Gamma_i$-fractional user preferences at the population level, and 2) that UP-AR also performs favorably on traditional evaluate metrics drawn from CARLA Pawelczyk et al. (2021). We used the native CARLA catalog for the `Give Me Some Credit (GMSC)` (Kaggle, 2011), `Adult Income (Adult)` (Dua & Graff, 2017) and `Correctional Offender Management Profiling for Alternative Sanctions (COMPAS)` (Angwin et al., 2016) data sets as well as pre-trained models (both the **Neural Network** (NN) and **Logistic Regression** (LR)). NN has three hidden layers of size [18, 9, 3], and the LR is a single input layer leading to a Softmax function. Although AR is proposed for *linear models*, it can be extended to *nonlinear models* by the local linear decision boundary approximation method LIME Ribeiro et al. (2016), which we refer to as AR-LIME.

**PERFORMANCE METRICS:** For UP-AR, we evaluate: 1) *Success Rate*: The percentage of adversely affected individuals for whom recourse was found. 2) *Average Time Taken*: The average time (in seconds) to generate recourse for a single individual. 3) *Constraint Violations*: The average number of non-actionable features modified. 4) *Redundancy*: A metric that tracks superfluous feature changes. For each successful recourse calculated on a univariate basis, features are flipped to their original value. The redundancy for recourse is the number of flips that do not change the model's classification decision. 5) *Proximity*: The normalized $l_2$ distance of recourse to its original point. 6) *Sparsity*: The average number of features modified.

We provide comparative results for UP-AR against state-of-the-art counterfactual/recourse generation techniques such as GS, Watcher, AR(-LIME), CCHAVE and FACE. These methods were selected based on their popularity and their representation of both independence and dependence based methods, as defined in CARLA. In addition to the traditional performance metrics, we also measure *Preference-Root mean squared error (pRMSE)* between the user preference score and the fractional cost of the suggested recourses. We calculate $pRMSE_i$ for a randomly selected continuous valued feature $i$ ~~using: We evaluate~~ $pRMSE_i$ ~~across the recourses generated for $n$ adversely affected individuals.~~

$$pRMSE_i = \sqrt{\frac{1}{n}\sum_{j=1}^{n}\left(\hat{\Gamma}_i^{(j)} - \Gamma_i^{(j)}\right)^2} \quad \text{where} \quad \hat{\Gamma}_i^{(j)} = \frac{cost\left(\mathbf{r}_i, \mathbf{x}_i\right)}{\sum_{k\in F_{con}} cost\left(\mathbf{r}_k, \mathbf{x}_k\right)} \tag{6}$$

Here $\Gamma_i^{(j)}$ and $\hat{\Gamma}_i^{(j)}$ are user provided and observed preference scores of feature $i$ for an individual $j$. **In Table 4, we summarize** $pRMSE$**, which is the average error across continuous features such that:** $pRMSE = \frac{1}{|F_{con}|}\sum_{i\in F_{con}} pRMSE_i$**.**

**DATASETS:** We train an LR model on the processed version of `german` Bache & Lichman (2013) credit dataset from *sklearn's linear_model* module. We replicate Ustun et al. (2019)'s model training and recourse generation on `german`. The dataset contains 1000 data points with 26 features for a loan application. The model decides if an applicant's credit request should be approved or not. Consider $F_{con} = \{LoanDuration, LoanAmount\}$ and $F_{cat} = \{CriticalAccountOrLoansElsewhere, HasGuarantor, HasCoapplicant\}$. Let the user scores for $F_{con}$ be $\Gamma = \{0.8, 0.2\}$ and ranking for $F_{cat}$ be $\{3, 1, 2\}$ for all the denied individuals. For this experiment, we set $\tau^{-1} = 4$. Out of 155 individuals with denied credit, AR and UP-AR provided recourses to 135 individuals.

**Cost Correction:** Out of all the denied individuals for whom categorical actions were suggested, an average of $\sim \$400$ in *LoanAmount* was recovered by cost correction.

Table 4: Summary of performance evaluation of UP-AR. Top performers are highlighted in green.

| Data. | Recourse Method | Neural Network Model | | | | | | | Logistic Regression | | | | | | |
|---|---|---|---|---|---|---|---|---|---|---|---|---|---|---|---|
| | | Succ Rate | pRM SE | Avg Tim. | Con. Vio. | Red. | Pro. | Spa. | Succ Rate | pRM SE | Avg Tim. | Con. Vio. | Red. | Pro. | Spa. |
| *GMSC* | GS | 0.75 | 0.16 | 0.02 | 0.00 | 6.95 | 1.01 | 8.89 | 0.62 | 0.18 | 0.03 | 0.00 | 4.08 | 1.39 | 8.99 |
| | Wachter | 1.00 | 0.18 | 0.02 | 1.49 | 6.84 | 1.08 | 8.46 | 1.00 | 0.17 | 0.03 | 1.23 | 3.51 | 1.42 | 7.18 |
| | AR(-LIME) | 0.03 | 0.17 | 0.45 | 0.00 | 0.00 | 0.17 | 1.72 | 0.17 | 0.17 | 0.73 | 0.00 | 0.00 | 0.93 | 1.91 |
| | CCHVAE | 1.00 | 0.18 | 1.05 | 2.0 | 9.99 | 1.15 | 10.1 | 1.00 | 0.18 | 1.37 | 2.00 | 8.64 | 2.05 | 11.0 |
| | FACE | 1.00 | 0.17 | 8.05 | 1.57 | 6.65 | 1.20 | 6.69 | 1.00 | 0.16 | 11.9 | 1.65 | 7.47 | 2.30 | 8.45 |
| | **UP-AR** | 0.94 | 0.07 | 0.08 | 0.00 | 1.30 | 0.49 | 3.22 | 1.00 | 0.07 | 0.12 | 0.00 | 1.47 | 0.68 | 3.92 |
| *Adult* | GS | 0.84 | 0.10 | 0.03 | 0.00 | 2.86 | 1.30 | 5.09 | 0.84 | 0.10 | 0.04 | 0.00 | 1.76 | 2.05 | 5.85 |
| | Wachter | 0.55 | 0.10 | 0.04 | 1.44 | 3.05 | 0.74 | 4.90 | 1.00 | 0.11 | 0.10 | 1.68 | 0.90 | 1.44 | 5.81 |
| | AR(-LIME) | 0.42 | 0.10 | 9.20 | 0.00 | 0.00 | 2.10 | 2.54 | 0.76 | 0.10 | 7.37 | 0.00 | 0.03 | 2.10 | 2.31 |
| | CCHVAE | 0.84 | 0.11 | 0.77 | 4.47 | 5.83 | 3.95 | 9.40 | 0.84 | 0.10 | 1.08 | 4.22 | 6.85 | 3.96 | 9.45 |
| | FACE | 1.00 | 0.10 | 6.78 | 4.58 | 7.54 | 4.11 | 7.91 | 1.00 | 0.10 | 8.37 | 4.53 | 5.91 | 4.28 | 7.81 |
| | **UP-AR** | 0.82 | 0.10 | 0.76 | 0.00 | 0.78 | 1.77 | 2.78 | 0.82 | 0.05 | 0.67 | 0.00 | 0.55 | 1.78 | 2.88 |
| *COMPAS* | GS | 1.00 | 0.15 | 0.03 | 0.00 | 1.09 | 0.47 | 3.35 | 1.00 | 0.14 | 0.04 | 0.00 | 0.34 | 1.12 | 3.98 |
| | Wachter | 1.00 | 0.14 | 0.05 | 1.00 | 1.61 | 0.56 | 4.35 | 1.00 | 0.14 | 0.04 | 1.00 | 0.85 | 1.06 | 4.83 |
| | AR(-LIME) | 0.65 | 0.13 | 0.20 | 0.00 | 0.00 | 0.78 | 0.90 | 0.52 | 0.15 | 0.24 | 0.00 | 0.00 | 1.45 | 1.57 |
| | CCHVAE | 1.00 | 0.14 | 5.09 | 2.27 | 4.31 | 1.70 | 4.91 | 1.00 | 0.14 | 0.02 | 1.62 | 2.70 | 1.74 | 4.92 |
| | FACE | 1.00 | 0.15 | 0.37 | 2.39 | 3.96 | 2.35 | 4.72 | 1.00 | 0.15 | 0.40 | 2.47 | 4.38 | 2.46 | 4.81 |
| | **UP-AR** | 0.92 | 0.08 | 0.04 | 0.00 | 0.60 | 0.63 | 1.82 | 1.00 | 0.10 | 0.05 | 0.00 | 0.81 | 0.82 | 2.74 |

For the following datasets, **for traditional metrics,** user preferences were set to be uniform for all actionable features to not bias the results to one feature preference over another: 1) **GMSC:** The data set from the 2011 Kaggle competition is a credit underwriting dataset with 11 features where the target is the presence of delinquency. Here, we measure what feature changes would lower the likelihood of delinquency. We again used the default protected features (*age* and *number of dependents*). The baseline accuracy for the NN model is 81%, while the baseline accuracy for the LR is 76%. 2) **Adult Income:** This dataset originates from 1994 census database with 14 attributes. The model decides whether an individual's income is higher than $50,000$ USD/year. The baseline accuracy for the NN model is 85%, while the baseline accuracy for the LR is 83%. Our experiment is conducted on a sample of 1000 data points. 3) **COMPAS:** The data set consists of 7 features describing offenders and a target representing predictions. Here, we measure what feature changes would change an automated recidivism prediction. The baseline accuracy for NN is 78%, while baseline accuracy for LR is 71%.

**PERFORMANCE ANALYSIS OF UP-AR:** We find UP-AR holistically performs favorably to its counterparts. Critically, it respects feature constraints (which we believe is fundamental to actionable recourse) while maintaining a significantly low redundancy and sparsity. This indicates that it tends to change fewer necessary features. Its speed makes it tractable for real-world use, while its proximity values show that it recovers relatively low-cost recourse. These results highlight the promise of UP-AR as a performative, low-cost option for calculating recourse when user preferences are paramount. UP-AR shows consistent improvements over all the performance metrics. The occasional lower success rate for a NN model is attributed to $0$ constraint violations.

$pRMSE$**: We analyze user preference performance in terms of $pRMSE$. From Table 4, we observe that UP-AR's $pRMSE$ is consistently better than the state of art recourse methods. The corresponding experimental details and visual representation of the distribution of $pRMSE$ is deferred to Appendix C.2.**

## 6 CONCLUDING REMARKS

In this study, we propose to capture different forms of user preferences and propose an optimization function to generate actionable recourse adhering to such constraints. We further provide an approach to generate a connected (Laugel et al., 2019) recourse guided by the user. We show how UP-AR adheres to soft constraints by evaluating user satisfaction in fractional cost ratio. We emphasize the need to capture various user preferences and communicate with the user in comprehensible form. This work motivates further research on how truthful reporting of preferences can help improve overall user satisfaction.

## 7 ETHICS STATEMENT

We proposed a recourse generation method for machine learning models that directly impact human lives. For practical purposes, we considered publicly available datasets for our experiments. Due care was taken not to induce any bias in this research. We further evaluated the primary performance metric for two sensitive groups (males and females) for *german* dataset.

This study reflects our efforts to bring human subjects within the framework of recourse generation. Comprehensible discussion with the users about the process improves trust and explainability of the steps taken during the entire mechanism. With machine learning models being deployed in high-impact societal applications, considering human inputs (in the form of preferences) for decision-making is a highly significant factor for improved trustworthiness. Additionally, comprehensible discussion with human subjects is another crucial component of our study. Our study motivates further research for capturing individual idiosyncrasies.

Gathering preferences from an individual could be another potential source of bias for UP-AR recourses, which needs to be evaluated with further research with human subjects. Preferential recourses will have a significant positive impact on humans conditioned on truthful reporting of various preferences. Preference scores are subject to various background factors affecting an individual, some of which can be sensitive. Additional care must be taken to provide confidentiality to these background factors while collecting individual preference scores, which have the potential to be exploited.

## 8 REPRODUCIBILITY STATEMENT

We have added the experiment and implementation details both in the paper and the supplementary materials to reproduce our reported results and our methodology. We have faithfully followed the proposed the algorithms to implement our solutions. We have clearly stated and cited the source of our data, as well as the evaluation metrics. We have also reported the computational time required to perform the experiments.

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

## A  APPENDIX

**We have added the following updates in the Appendix to the latest document**

1. **Section B on User acceptance survey.**
2. **Section C.2 on Random user preference study.**
3. **Section C.7 on UP-AR scalability with actionable features dimension.**

### A.1  ANALYSIS

**Interpretable and Incremental steps:**  In this study, each step $\delta_i^{(t)}$ is a predefined minimal feature modification inherently derived from the feature vector $\mathbf{x}$. A recourse suggested by UP-AR can be broken down into interpretable actions. Alice was denied a loan application, and her suggested recourse is to decrease the loan amount from \$8072 to \$6472 and decrease the loan duration from 30 years to 10 years. Here the recourse is broken down into reducing the loan amount by 16 steps of \$100 each, implying that the loan amount is 16 steps connected with the original feature value. Such steps increase the comprehensibility of recourse.

## B  USER ACCEPTANCE SURVEY

**We conducted a survey with** $40$ **random students and employees from a mailing list. The survey included one question with four options as follows:** *If you are denied a loan application. What do you expect from the bank to get your loan approved ?*

1. *A single list of suggestions to your profile. Example: (increase income by 100\$ and reduce loan duration by 1 year)*
2. *A set with multiple lists of suggestions to your profile. Example: (i) increase income by 100\$ and reduce loan duration by 1 year OR ii) increase income by 500\$ OR iii) reduce loan duration by 3 year OR iv) bring a co-applicant)*
3. *Influence the bank's suggestions by providing preferential scores for actions you can take. Example: (preferring to increase loan duration more than loan amount by 8:2, or preferring to bring a guarantor before a co-applicant)*
4. *Any other form of preferences*

**Every individual in the survey was asked to select one of the four choices provided. In this survey, it is identified that majority of** $60\%$ **of individuals preferred influencing the bank's decision by providing preference scores for individual features, followed by** $30\%$ **of individuals who wanted multiple recourses from the bank. The remaining** $10\%$ **of individuals preferred a single recourse or any other form of preference.**

## C  ABLATION STUDIES

In this section, we perform multiple experiments to understand several properties of UP-AR. First, run an experiment to measure the disparities in $pRMSE$ between the two gender groups. Secondly, we run experiments to understand the effects of the temperature parameter $\tau$ on UP-AR. Thirdly, we try to understand the relation between $T^*$ and $\hat{\Gamma}$, if any.

### C.1  UP-AR USER PREFERENCE DISPARITIES

UP-AR satisfies user Type I user preferences as observed in Section 4. For the following experiment, we consider a similar setup as in Section 4. We now evaluate similar performance among *males* and *females* separately in terms of $pRMSE$. With a similar setup as Section 4, Figure 6, shows a distribution of cost between the two gender groups. Observed $pRMSE_{LoanDuration}$ for males is 0.09, whereas for females it is 0.11. With this simple experiment, we conclude that UP-AR does not show any significant disparities in terms of adhering to user preferences.

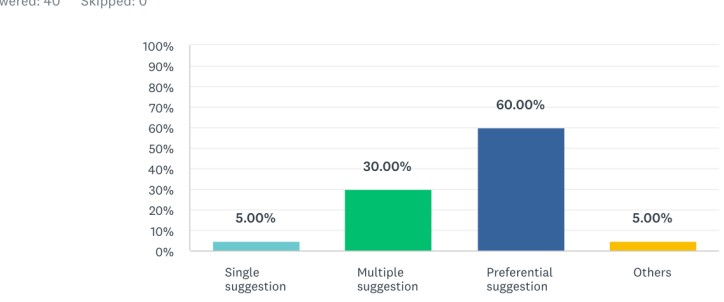

If you are denied a loan application. What do you expect from the bank to get your loan approved ?

Answered: 40     Skipped: 0

| ANSWER CHOICES | RESPONSES | |
|---|---|---|
| ▼ Single suggestion | 5.00% | 2 |
| ▼ Multiple suggestion | 30.00% | 12 |
| ▼ Preferential suggestion | 60.00% | 24 |
| ▼ Others | 5.00% | 2 |
| TOTAL | | 40 |

Figure 5: Snapshot of the human acceptance survey.

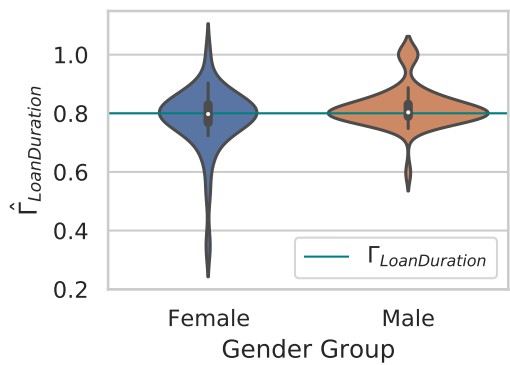

Figure 6: Comparison of UP-AR's distribution of $\hat{\Gamma}_{LoanDuration}$ between males and females for a *Logistic Regression* model trained on *German*.

## C.2 RANDOM USER PREFERENCE STUDY

We performed an experiment with increasing step sizes on *German* dataset with similar experimental setup as in previous section. We observed that, with increasing step sizes, $pRMSE_i$ increased from $0.09$ to $0.13$, whereas it was consistent for AR.

In the next experiment, we randomly choose user preference for *LoanDuration* from $[0.4, 0.5, 0.6, 0.7, 0.8]$. The rest of the experimental setup is identical to the setup discussed in Section 4. In this experiment, we observe $pRMSE$ with non-universal user preference for adversely affected individuals. Here the average $pRMSE$ of both *LoanDuration* and *LoadAmount* for UP-AR is $0.19$, whereas for AR it is $0.34$.

Further, using the CARLA package, we generated recourses for a set of $1000$ individuals and $\Gamma$ for two continuous features was randomly selected from $[0.3, 0.6, 0.9]$. Figure 9 provides a visual analysis of the distribution of average $pRMSE$ using violin plots. The

experiments were performed on the $3$ datasets discussed in Section 5 for both the LR and NN models. For *GMSC* dataset, $F_{con}$ = {*DebtRatio, MonthlyIncome*} and $F_A$ = {*RevolvingUtilizationOfUnsecuredLines, NumberOfTime30-59DaysPastDueNotWorse, DebtRatio, MonthlyIncome, NumberOfOpenCreditLinesAndLoans, NumberOfTimes90DaysLate, NumberRealEstateLoansOrLines, NumberOfTime60-89DaysPastDueNotWorse*}. For *COMPAS* dataset, $F_{con}$ = {*priors-count, length-of-stay*} and $F_A$ = {*two-year-recid, priors-count' length-of-stay*}. For *Adult* dataset, $F_{con}$ = {*education-num, capital-gain*} and $F_A$ = {*education-num, capital-gain, capital-loss, hours-per-week, workclass-Non-Private, workclass-Private, marital-status-Married, marital-status-Non-Married, occupation-Managerial-Specialist, occupation-Other*}.

With these experiments we conclude that UP-AR's $\hat{\Gamma}$ deviation from the user's $\Gamma$ is consistently lower than the existing recourse generation methodologies. We observe that AR is unaffected by the varying user preference due to the fact that AR and other state-of-the-art recourse methodologies lack the capability of capturing such idiosyncrasies. On the other hand, UP-AR is driven by those preferences and has significantly better $pRMSE$ in comparison to AR.

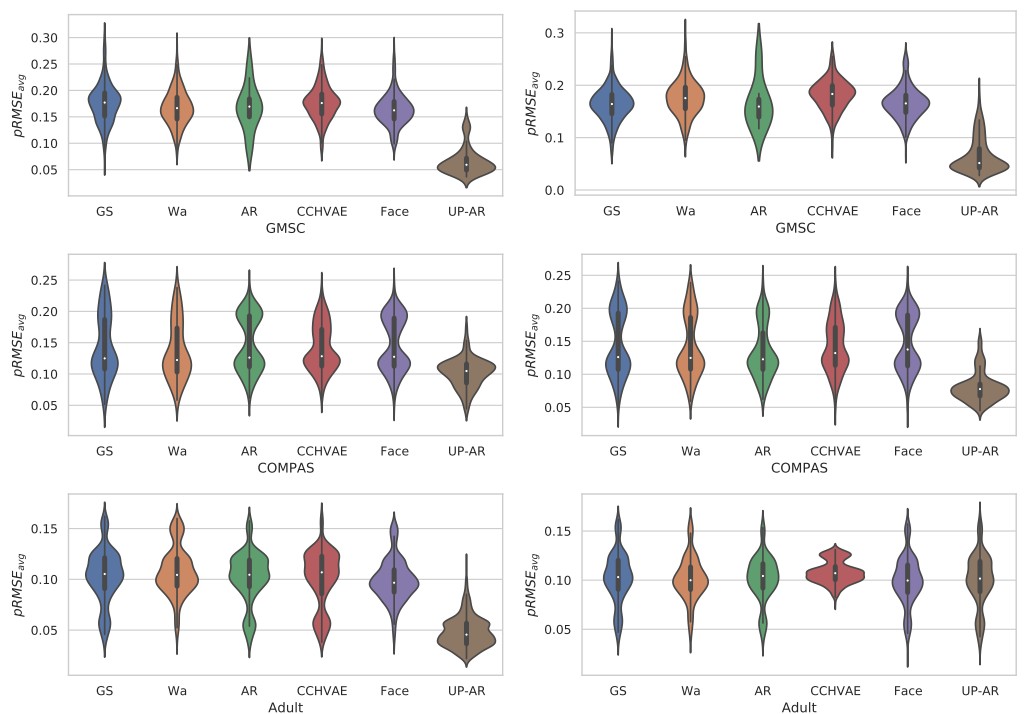

Figure 7: Logistic Regression model          Figure 8: Neural Network model

Figure 9: Distribution of the average $pRMSE$ of UP-AR and other recourse methodologies.

### C.3 Ablation study on $\tau$

For the following experiment, we again consider a similar setup as in Section 4. Each data point in the plot represents the mean total cost of recourses for the target population for $20$ independent runs of UP-AR, and the shaded region represents the $\pm 1$ standard deviation of the $20$ runs. We observe:

1. Effect of calibrating the overall cost of target population using $\tau$. $\tau$ controls the frequency of categorical actions detailed in Section 3.1.1.

2. $\hat{\Gamma}_{LoanDuration}$ is not affected by any setting of $\tau$ as observed in Figure 11.

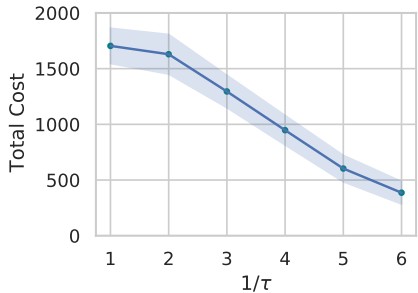 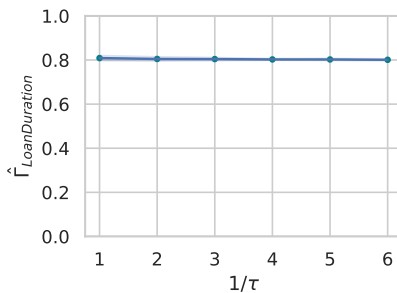

Figure 10: Total cost of the recourses generated for target population for varying $\tau$. The user preference scores are fixed for the individuals.

Figure 11: Mean fractional feature cost ratio of *LoanDuration* for varying $\tau$. For this experiment, $\Gamma_{LoanDuration}$ is set to $0.8$ for the target population.

### C.4 RELATION BETWEEN $\hat{\Gamma}$ AND $T^*$

Again considering a similar setup as in Section 4, Figure 12 visualizes the relation between the observed $\hat{\Gamma}_{LoanDuration}$ and the number of steps taken to identify a recourse $T^*$. We conclude that $\hat{\Gamma}_{LoanDuration}$ is not affected by the number of steps taken to identify a recourse by UP-AR.

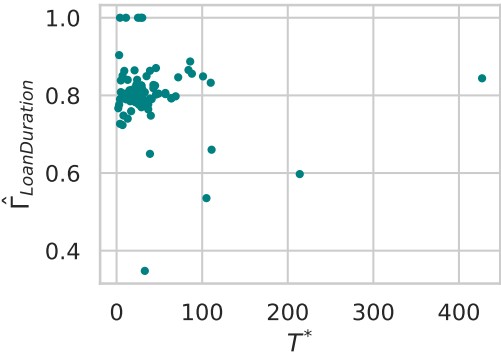

Figure 12: Scatter plot between $\hat{\Gamma}_{LoanDuration}$ and $T^*$ on the recourses generated for adversely affected target population.

### C.5 REAL COST VS EXPECTED COST

In this experiment, we compare the expected cost and the actual observed cost of the recourses generated. Figure 13 visualizes the expected cost and observed cost for actionable features. We observe that with increasing $\tau$, the total cost of recourses increases suggesting high categorical actions suggested in the generated recourses. Additionally, We also notice the consistency in $\hat{\Gamma}_{LoanDuration}$ for varying $\tau$. Please note that careful calibration of $\tau$ can help individuals who prefer categorical feature actions over continuous features.

### C.6 ABLATION STUDY ON CC

In Table 5 we explore the effect of UP-AR's cost correction procedure on the Adult and COMPAS datasets. We do not include the GMSC dataset as it does not include binary features, and therefore does not utilize the cost correction procedure. In Table 5 we show the number of factuals, the percentage of factuals for which recourse was found, the percentage of recourse found which contained at least one binary action, the percent of recourse found which underwent cost correction, the average percentage of steps saved by the cost correction procedure, and the average percent of cost

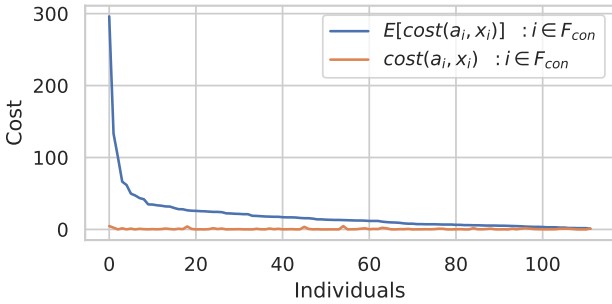

Figure 13: Expected and observed cost of modifications on $F_{con}$ for all the recourses generated on the adversely affected target population.

savings, measured as the percent reduction in continuous cost ($l_2$ distance) between a factual and its recourse before and after the cost-correction procedure.

Table 5: The Frequency and Effect of Cost Correction

| Metrics | Adult | COMPAS |
|---|---|---|
| Number of Factuals | 1000 | 568 |
| Success Rate | 79.3% | 99.6% |
| Percent of Recourse with a Binary Action | 71.9% | 82.6% |
| Percent of Recourse with Cost Correction | 38.4% | 25.5% |
| Average Percentage of Steps Saved | 67.9% | 63.5% |
| Average Percentage of Continuous Cost Saved | 83.1% | 76.0% |

### C.7 ABLATION STUDY ON ACTIONABLE FEATURE SET

We conducted an experiment on the average computational cost (modeled by execution time) of UP-AR and GS across a varying number of actionable features to explore how their performance changes as the actionable set size increases. Figures 14 and 15 show the performance trends for an *LR* model and *NN* model on the *Adult Income* dataset, while figures 16 and 17 show the performance trends for an *LR* model and *NN* model on the *German Credit* dataset.

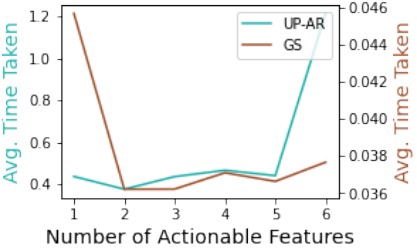
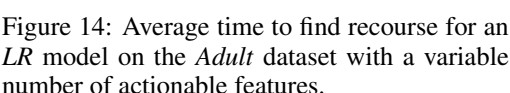
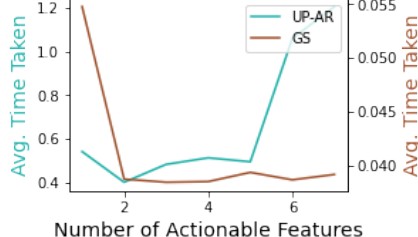

Figure 14: Average time to find recourse for an *LR* model on the *Adult* dataset with a variable number of actionable features.

Figure 15: Average time to find recourse for a *NN* on the *Adult* dataset with a variable number of actionable features.

We observe that UP-AR's average time increases as the actionable feature dimension increases whereas gradient based GS remains relatively consistent. This can be attributed to the additional user scoring preference and ranking preference constraints while identifying a recourse, as well as the cost correction procedure as the number of binary changes increases.

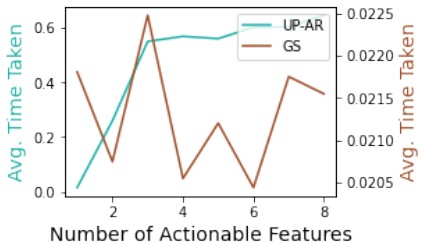 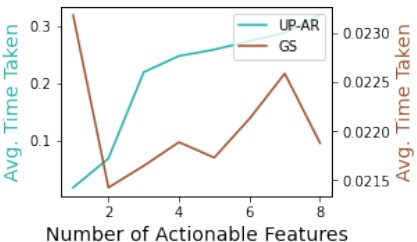

Figure 16: Average time to find recourse for an *LR* model on the *Credit* dataset with a variable number of actionable features.

Figure 17: Average time to find recourse for a *NN* on the *Credit* dataset with a variable number of actionable features.

## C.8 Additional proofs of results discussed in Section 4

### C.8.1 Proof of Lemma 1

Consider that recourse $\mathbf{r}$ was suggested by UP-AR for Alice represented by a feature vector $\mathbf{x}$. Let $\mathbf{r}$ was obtained at time step $T^*$. Here $cost\left(\mathbf{r}_i^{(t)}, \mathbf{x}_i + \mathbf{r}_i^{(t-1)}\right)$ measures the cost of taking an action $\mathbf{r}_i^{(t)}$ at time $t-1$ for feature $i$.

$$\mathbb{E}\left[cost\left(\mathbf{r}_i, \mathbf{x}_i\right)\right] = \mathbb{E}\left[\sum_{t=1}^{T} I_i^{(t)} cost\left(\mathbf{r}_i^{(t)}, \mathbf{x}_i + \mathbf{r}_i^{(t-1)}\right)\right]$$

$$= \sum_{t=1}^{T} \mathbb{E}\left[I_i^{(t)}\right] cost\left(\mathbf{r}_i^{(t)}, \mathbf{x}_i + \mathbf{r}_i^{(t-1)}\right)$$

$$= \sum_{t=1}^{T} \mathbb{P}\left[I_i^{(t)}\right] cost\left(\mathbf{r}_i^{(t)}, \mathbf{x}_i + \mathbf{r}_i^{(t-1)}\right)$$

Steps for each feature action at time $t$ are decided by the inverse cost weighted by user preference score $\Gamma_i$. Let us call this *weighted inverse cost* which is then mapped to a probability distribution using usual choices such as normalization or a softmax function. Let $\sigma(\cdot)$ be a function which maps *weighted inverse cost* to a probability distribution. We have,

$$\mathbb{E}\left[cost\left(\mathbf{r}_i, \mathbf{x}_i\right)\right] = \sum_{t=1}^{T} \sigma\left(\frac{\Gamma_i}{cost\left(\mathbf{r}_i^{(t)}, \mathbf{x}_i + \mathbf{r}_i^{(t-1)}\right)}\right) cost\left(\mathbf{r}_i^{(t)}, \mathbf{x}_i + \mathbf{r}_i^{(t-1)}\right)$$

With $C_{i,min}^{(T^*)}$ and $C_{i,max}^{(T^*)}$ representing the minimum and maximum cost of any step for feature $i$ until time step $T^*$, we obtain the result in Lemma 1.

### C.8.2 Proof of Corollary 1

For simplicity, consider a cost function where the overall cost of recourse is the sum total of individual feature action costs, i.e., $cost\left(\mathbf{r}, \mathbf{x}\right) = \sum_{i \in F_A} cost\left(\mathbf{r}_i, \mathbf{x}_i\right)$. The total expected cost of a recourse $\mathbf{r}$ is:

$$\mathbb{E}\left[cost\left(\mathbf{r}, \mathbf{x}\right)\right] = \sum_{t \in T^*} \sum_{i \in F_A} \sigma\left(\frac{\Gamma_i}{cost\left(\mathbf{r}_i^{(t)}, \mathbf{x}_i + \mathbf{r}_i^{(t-1)}\right)}\right) cost\left(\mathbf{r}_i^{(t)}, \mathbf{x}_i + \mathbf{r}_i^{(t-1)}\right)$$

Considering that all the steps are of equal cost and a linear function $\sigma(\cdot)$, we get Corollary 1:

$$\mathbb{E}\left[cost\left(\mathbf{r}, \mathbf{x}\right)\right] = T^* \sum_{i \in F_A} \sigma\left(\Gamma_i\right)$$

