# OpenReview forum: "Actionable Recourse Guided by User Preference"
_ICLR.cc/2023/Conference — Submitted to ICLR 2023_

### Official Review · Reviewer_uWp4 · 2022-10-24

**Confidence:** 4
**Correctness:** 2
**Technical Novelty And Significance:** 2
**Empirical Novelty And Significance:** 2
**Recommendation:** 3

**Clarity, Quality, Novelty And Reproducibility:**

The paper is well organized and clear to follow.
A minor comment: Certain figures are only described (for example Figures 3 and 4) but not discussed in greater detail. Readers will benefit with a detailed discussion on these results.


**Strength And Weaknesses:**

Strengths

Generating recourses that satisfy user-preference is an important research problem. While the cost function is shared across all the users, the authors propose a novel way to define a user’s preference in terms of contribution of a single feature to the overall cost.

The proposed technique is sound and quite easy to understand. The authors show that the expected cost of a feature is proportional to the user-preference.

Experiments on multiple datasets indicate that the proposed method is capable of generating recourses with minimal constraint violations.

Weaknesses

The fractional-score based user-preference, although novel, may not be as simple and easily expressible as proposed, especially, when the number of continuous features is large. Moreover, the preference input process puts an extra cognitive burden on the users. Evidence supporting the acceptability of such a preference elicitation process will go a long way in strengthening the paper. Separating continuous and categorical features may result in a manageable modeling task, but I wonder if a user would find it difficult to separate the two sets of features while providing the preferences.

How are the continuous features discretized? Does this have a bearing on the algorithm?

I recognize the challenges in modeling heterogeneous features (categorical and continuous). However, I am not convinced with the automatic assumption that the fractional cost of categorical features changes should be higher than that of continuous features (for all categorical features listed in the user’s preference rank, $ \Gamma_i = 1 $). A user might actually prefer changes to continuous features over categorical features. How does UP-AR handle such situations?

While the proposed method ensures that the expected cost follows the user-preference, I am unsure how this translates into practice. A user would expect a single (or at best a few diverse) recourse(s) satisfying the indicated preferences. Thus, usability of UP-AR is significantly hampered, as the preference compatibility of a single recourse cannot be guaranteed.

I am also concerned about the scanty experiments on preference-enforcement. The authors’ have shown pRMSE results only for single features (LoanDuration and DebtRatio). Evidence on other features, perhaps an average over all continuous features could have been reported for better insights into the empirical effectiveness of UP-AR.


**Summary Of The Paper:**

Most recourse generation methods optimize for properties such as proximity, sparsity, validity, etc, but often ignore the user’s preference. Recent effort in taking into account the individual preference requires a shared cost function on feature changes or user-specified feature change costs that can be burdensome on the users. The authors propose to capture user preferences in terms of user-specified- scoring for continuous features, bounds on feature values, and ranking of categorical attributes. The fractional scores on continuous attributes allow the user to indicate the contribution of each feature on the overall cost. These preferences result in a customized optimization problem that is solved by the UP-AR algorithm. The UP-AR algorithm is a coordinate wise gradient descent/ascent where the step size is influenced by the user-specified cost. The authors also propose a roll-back mechanism to reverse changes to features that are unnecessary due to a change to a categorical attribute. Experiments conducted on benchmarking datasets indicate the effectiveness of the proposed approach.


**Summary Of The Review:**

Overall, the paper has interesting ideas that might help to push the frontiers of algorithmic recourse research. However, I have major concerns related to the preference definition, guarantees of the algorithm and lack of experiments.

---

> ### Author Response · Authors · 2022-11-13
> **We thank the reviewers for their valuable feedback. Here are our responses to the concerns raised:**
>
> **The fractional-score based user-preference, although novel, may not be as simple and easily expressible as proposed, especially, when the number of continuous features is large. Moreover, the preference input process puts an extra cognitive burden on the users. Evidence supporting the acceptability of such a preference elicitation process will go a long way in strengthening the paper.**
>
> *Response:* Gathering actual individual cost $userCost(r_i,x_i)$ of a feature action is naturally difficult. A user may not specifically communicate her personal costs of a feature action. We would like to point out the inherent complexity in soliciting such individual cost information. Hence, we fall back to gathering preference scores $\Gamma_{i}$ for the fractional cost, which individuals prefer, as observed in our survey. We acknowledge the difficulty of scoring for a large number of continuous features. However, a user has the flexibility to provide actionability of a feature and can bring down the cardinality of the continuous feature set since **sparsity** is a highly desirable property of a recourse. A study of the acceptability of such scores is added as Appendix B in the updated document. Our anonymous survey with 40 individuals concluded that approximately 60\% of individuals prefer providing preference scores for actionable features, followed by 30\% individuals who preferred multiple recourses to choose from. The rest of the subjects either preferred a single recourse or any other form of preference solicitation. We thank the reviewers for this recommendation.
>
> **Separating continuous and categorical features may result in a manageable modeling task, but I wonder if a user would find it difficult to separate the two sets of features while providing the preferences.**
>
> *Response:* As suggested in Section 2.1, a user will be provided with two sets of continuous features $F_{con}$ for scoring and categorical $F_{cat}$ for ranking. Please note that we do not imagine we will offload the task to users, but rather the model provider is responsible for separating the features and collecting required scores from the applicant.
>
> **How are the continuous features discretized? Does this have a bearing on the algorithm?**
>
> *Response:* Discretization of features is discussed in Section 2.1 and for this study continuous features are discretized with uniform step sizes —the smaller the steps, the better the UP-AR's adaptability to user preference. With larger steps, low-cost recourses are obtained with minimal steps, and adherence to user preference scores $\Gamma_{i}, \forall~i\in F_{con}$ for continuous features may not be guaranteed, given the stochastic nature of the recourse generation methodology. Please remember that the step sizes $\delta{}$ for a feature can be non uniform and predetermined by a domain expert. We performed an experiment with increasing step sizes on *German* dataset. We observed that, with increasing step sizes, $pRMSE_i$ increased from $0.09$ to $0.13$, whereas it was consistent for AR and these results are included in Appendix C.2.
>
>
> **I recognize the challenges in modeling heterogeneous features (categorical and continuous). However, I am not convinced with the automatic assumption that the fractional cost of categorical features changes should be higher than that of continuous features (for all categorical features listed in the user’s preference rank). How does UP-AR handle such situations?**
>
> *Response:* As discussed in Section~3.1.1, the frequency of categorical actions can be controlled by $\tau$, and if a user prefers categorical actions to continuous, careful calibration of $\tau$ can tune the UP-AR to adhere to such preferences. A higher value for $\tau$ tunes UP-AR into suggesting categorical actions more often. Figure 10 in the Appendix gives a quick visualization of the effect of $\tau$ on UP-AR.

---

> > ### Author Response · Authors · 2022-11-13
> > **Continued...**
> >
> > **While the proposed method ensures that the expected cost follows the user-preference, I am unsure how this translates into practice. A user would expect a single (or at best a few diverse) recourse(s) satisfying the indicated preferences. Thus, usability of UP-AR is significantly hampered, as the preference compatibility of a single recourse cannot be guaranteed.**
> >
> >
> > *Response:* Type II and Type III user preferences are enforced by hard constraints during the UP-AR's recourse generation strategy. Corollary 1 shows that Type I user preference is guaranteed in expectation. This opens the possibility of varying the parameters for the other type of features to offer a diverse set of recourses. UP-AR has the capability of providing diverse set of recourses where Type II and Type III preferences are always guaranteed and we acknowledge the limitation of observing some variance in $\hat{\Gamma_{}}$ for multiple recourses. We refer the reviewer to Table 3, where different preferences lead to unique recourses for an individual. UP-AR has sufficient capabilities for providing a diverse set of recourses such as: i) Updating the actionable features set, ii) calibrating the frequency of categorical actions using $\tau$, and iii) gathering multiple preferences from the user or suggesting recourses based on popular preference choices. By employing these available techniques, UP-AR can support a diverse set of recourses for an individual. However, we believe that UP-AR provides the most preferred recourse for an individual, which satisfies the three types of preference scores provided and has a higher chance of acceptability.
> >
> >
> > **I am also concerned about the scanty experiments on preference-enforcement. The authors’ have shown pRMSE results only for single features (LoanDuration and DebtRatio). Evidence on other features, perhaps an average over all continuous features could have been reported for better insights into the empirical effectiveness of UP-AR.**
> >
> > *Response:* We have performed the average pRMSE evaluation with random user preferences across datasets and included them in the main evaluation Table as a part of the performance metrics. Additional violin plots with the distribution of pRMSE are also included in the Appendix C.2.  Additionally, Figure 3 and Figure 4 also show the distribution of $\hat{\Gamma_{i}}$ for both linear model and a neural network model, which we observe to be centered around $\Gamma_{i}$. Further, we have performed the following experiments with *German* data and included them in Appendix C.2.
> > 1) With the same universal user preference setup, the average $pRMSE$ of both *LoanDuration* and *LoadAmount* for UP-AR is $0.10$, whereas for AR it is $0.34$.
> > 2) With randomly choosing user preference for \textit{LoanDuration} from $[0.4, 0.5, 0.6, 0.7, 0.8]$. Here the average $pRMSE$ of both *LoanDuration* and *LoadAmount* for UP-AR is $0.19$, whereas for AR it is $0.34$. We have included these findings in the Appendix C.2.
> >
> > Additionally, we have also considered gender disparity study of UP-AR in Appendix C.1 and concluded its consistency of $pRMSE$ across males and females. The results are visualized in Figure 6 of the Appendix. Please note that these additional experiments were differed to the Appendix due to the space limitation in the main document.
> >
> > **Certain figures are only described (for example Figures 3 and 4) but not discussed in greater detail. Readers will benefit with a detailed discussion on these results.**
> >
> > *Response:* We thank the reviewers for the suggestion. We have included: *These experiments signify the adaptability of the UP-AR to user preferences and provides empirical evidence that UP-AR's distribution of $\hat{\Gamma_{i}}$ is centered around $\Gamma_{i}$.* in Section 4 of the updated document.

---

> > > ### Author Response · Authors · 2022-12-03
> > > **gentle follow up**
> > >
> > > Dear Reviewer uWp4,
> > >
> > > Thank you again for your review comments. We have provided detailed responses. We'd like to follow up to see if there is anything else that we could further clarify. Thank you.
> > >
> > > Best,
> > >
> > > Authors

---

> ### Author Response · Authors · 2022-11-18
> **pRMSE added to the main evaluation table**
>
> As per the recommendations, we have performed the average pRMSE evaluation across datasets and included them in the main evaluation Table as a part of the performance metrics. Additional violin plots with the distribution of pRMSE are also included in the appendix. We thank the reviewers for this suggestion.

---

### Official Review · Reviewer_zS7o · 2022-10-25

**Confidence:** 3
**Correctness:** 3
**Technical Novelty And Significance:** 3
**Empirical Novelty And Significance:** 2
**Recommendation:** 6

**Clarity, Quality, Novelty And Reproducibility:**

Clarity:

This paper is well-written and easy to read.

Quality and Novelty:

This paper studies a novel problem, which incorporates individual user-level preference into the optimization of the actionable resource. The proposed method is well-motivated, solid and shows strong empirical performance.

Reproducibility:
All the experiment details and implementations are provided.

**Strength And Weaknesses:**

Strength:

<1>. This paper is very well-written and well-motivated. I enjoy reading the paper and learning from it.

<2>. The problem setup it studied is novel, which considers the usage of user-preference in actionable resource optimization, and it is closely related with lots of real-world scenarios.

<3>. The proposed method is solid, easy to implement, and shows great empirical performance.

Weakness:

<1>. The main argument of the paper is the individual user-level performance, and the proposed method is tailored to this. However, in the experiments, if i understand it correctly, it seems the set-up uses the universal user preference function? This might limit the argument from the empirical experiments. Could the authors comment more about this?

<2>. The current empirical set-up for UP-AR is based on linear models, could the authors comment more about the feasibility of scaling it to more complicated neural networks?

<3>. Another aspect is the actionable feature dimension, how the proposed method scale with that?

<4>. Getting user preference might be tricky in real-world settings, could we utilize any signal from training data to get this information? Could the authors comment more on this?

**Summary Of The Paper:**

This paper studies how to generate actionable resources for different individuals, with their specific individual user-based preferences. Specifically, it discusses three types/formats of user preferences: (1). Scoring for continuous features; (2). Ranking for categorical features as well as (3). Bounding for feature values. Based on these three types of constraints, it proposes an optimization framework which minimizes the cost of the overall change. It further instantiated the method with a UP-AR algorithm, which consists of two stages: candidate generation using gradient-descent, followed by the redundancy and cost correction. Empirical results also show superior performance compared with baseline.

**Summary Of The Review:**

This paper is a nice paper in studying effective methods to incorporate user-level preference into the actionable resource optimization problem. The proposed method is novel and well-motivated. Empirically, it also shows strong performance compared with baselines. However, there are some issues in the current set-up of the experiment, such as the universal user-preference, the availability of the user-preference, etc.

---

> ### Author Response · Authors · 2022-11-13
> **We thank the reviewers for their valuable feedback. Here are our responses to the concerns raised:**
>
> **The main argument of the paper is the individual user-level performance, and the proposed method is tailored to this. However, in the experiments, if i understand it correctly, it seems the set-up uses the universal user preference function? This might limit the argument from the empirical experiments. Could the authors comment more about this?**
>
> *Response:* Identifying preferred actionable recourse for one individual is independent of any other individual. Universal preference setup for most of the experiments is considered solely for baseline comparisons. Building on this observation, recourses consolidated in Table 3 are identified for an individual with different preferences, and this table should shed light on UP-AR’s adaptability to preferences. We have performed the average pRMSE evaluation with random user preferences across datasets and included them in the main evaluation Table as a part of the performance metrics. Additional violin plots with the distribution of pRMSE are also included in the Appendix C.2. We thank the reviewers for this suggestion. We have additionally performed another experiment with *German* data with randomly choosing user preference for *LoanDuration* from $[0.4, 0.5, 0.6, 0.7, 0.8]$. Here the average $pRMSE$ of both *LoanDuration* and *LoadAmount* for UP-AR is $0.19$, whereas for AR it is $0.34$. We thank the reviewers for the comments and these findings are also included in the Appendix C.2.
>
> **The current empirical set-up for UP-AR is based on linear models, could the authors comment more about the feasibility of scaling it to more complicated neural networks?**
>
> *Response:* Although our running experiments and ablation studies were performed using a Logistic Regression model, we have also performed experiments to evaluate traditional metrics using a **neural network** model in Table 4. pRMSE is also added as a performance metric in the main evaluation table. Figure 3 and Figure 4 also show the distribution of $\hat{\Gamma_{i}}$ for both linear model and a neural network model, which we observe to be centered around $\Gamma_{i}$.
>
> **Another aspect is the actionable feature dimension, how the proposed method scale with that?**
>
> *Response:* As the number of actionable features increases, UP-AR's actionability space increases. This observation is consistent with all the other baseline recourse generation methodologies. Additionally, we have included the results on how UP-AR's time scales with number of actionable features in the Appendix C.7 of the updated document. It is observed that UP-AR's average time increases as the actionable feature dimension increases whereas gradient based Growing Spheres remains consistent. This can be attributed to the additional **user scoring preference, ranking preference constraints and the cost correction procedure** while identifying a recourse. However, please note the importance of **Sparsity** for an actionable recourse which states that a recourse with fewer actionable features is preferred.
>
> **Getting user preference might be tricky in real-world settings, could we utilize any signal from training data to get this information? Could the authors comment more on this?**
>
> *Response:* We would like to stress upon the crucial idiosyncrasies between two identical individuals and difficulty of an individual to elicit their actual cost of taking an action. These preferences are highly influenced by several factors such as an **individuals socioeconomic status or emotional aspects** etc. We believe any preferential signal from the training dataset could bias the recourse generation methodology. Additionally, we would like to point out that this study aims to identify an actionable recourse for individuals by soliciting specific preference scores $\Gamma_{i}$ as a proxy for feature-specific costs $userCost(r_i,x_i)$. Additionally, if individuals cannot specify their preferences, UP-AR can fallback uniform preference scores $(\Gamma_{i}=\Gamma_{j}, \forall~i,j \in F_A)$ for the features for suggesting a recourse which is the existing state of the art. Actionable Recourse (AR) paper was motivated by characterizing the difficulty of moving a prediction by considering quantile function as a proxy to the universal cost function. However, soliciting personal preference will remove possible biases in these universal modelings.

---

### Official Review · Reviewer_7UMN · 2022-10-27

**Confidence:** 3
**Correctness:** 3
**Technical Novelty And Significance:** 3
**Empirical Novelty And Significance:** 3
**Recommendation:** 6

**Clarity, Quality, Novelty And Reproducibility:**

This work is well written with good clarity, and the presented method is interesting to me with technical novelty. The proposed method is also shown effective empirically.

**Strength And Weaknesses:**

Strength: This work provides three types of user preferences and adopt them as soft constraints into the actionable recourse formulation. This idea is interesting and has shown helpful in generating actionable recourse. A two-stage UP-AR and some theoretical analysis are also provided to consolidate this work.

Weakness:
1) In Tab 5, though UP-AR performs well in terms of indices such as Sparsity and Redundancy, the success rate is not always the highest among baselines. For example, on the Adult dataset, its success rate is even much worse than other methods. I am wondering whether it’s fair to make a comparison among different methods under this setting.
2) This work adopts the logistic regression model during experiments evaluation, while more sophisticated models remain unexplored.
3) Though user preference is interesting, I’m not clear whether such information is always available to be used. This may limit the applicability of the method.


**Summary Of The Paper:**

This work studies Actionable Recourse (AR) and proposes to incorporate user preferences as constraints into the recourse generation process. The authors propose three forms of user preferences: scoring continuous features, bounding feature values, and ranking categorical features. An optimization approach is provided to find the recourse. Numerical comparisons were conducted to evaluate the proposed method.

**Summary Of The Review:**

This work proposes an interesting idea that models three user preferences for actionable recourse and designs an optimization method to generate the recourse. Some analysis is given and experiments were conducted on 2 datasets to verify the effectiveness. Nevertheless, I have some concerns (see Weakness) that need to be addressed by the authors.

---

> ### Author Response · Authors · 2022-11-13
> **We thank the reviewers for their valuable feedback. Here are our responses to the concerns raised:**
>
> **In Tab 5, though UP-AR performs well in terms of indices such as Sparsity and Redundancy, the success rate is not always the highest among baselines. For example, on the Adult dataset, its success rate is even much worse than other methods. I am wondering whether it’s fair to make a comparison among different methods under this setting.**
>
> *Response:* We acknowledge the lower success rate of UP-AR for the Adult dataset. However, this can be attributed to 0 constraint violations while finding a recourse. With these criteria, a fair comparison would be with GS and AR(-LIME) in the table, and as observed, the success rate of UP-AR is similar to these methods. We considered defining success only as a recourse that doesn't violate constraints, but this yielded a ~0\% success rate for some of the popular methods we compare to. We find that the convention in the literature is to report success rate alongside constraint violations so both can be understood in context, so we elected to conform to that standard.
>
> **This work adopts the logistic regression model during experiments evaluation, while more sophisticated models remain unexplored.**
>
> *Response:* We have included $pRMSE$ as a metric in the main evaluation table. Additionally, Figure 3 and Figure 4 also show the distribution of $\hat{\Gamma_{i}}$ for both linear model and a neural network model, which we observe to be centered around $\Gamma_{i}$. Additionally, we considered logistic regression for ablation studies, while for Table 4, with the traditional performance metrics, our experiments include **neural network model** consistent with prior studies in this field and the CARLA bench-marking tool.
>
> **Though user preference is interesting, I’m not clear whether such information is always available to be used. This may limit the applicability of the method.**
>
> *Response:* Gathering actual individual cost of a feature action is naturally difficult. A user may not specifically communicate the costs ($userCost(r_i,x_i)$) of a feature action $r_i$. We would like to point out the inherent complexity in soliciting such individual cost information. Hence, we fall back to gathering preference scores $\Gamma_{i}$ for the fractional cost, which individuals prefer, as observed in our survey. With this constraint, we strongly believe our setting is straightforward for an applicant to comprehend. Additionally, if individuals cannot specify their preferences, UP-AR can fallback to uniform preference scores $(\Gamma_{i}=\Gamma_{j}, \forall~i,j \in F_A)$ for the features for suggesting a recourse which is the existing state of the art. Actionable Recourse (AR) paper was motivated by characterizing the difficulty of moving a prediction by considering quantile function as a proxy to the universal cost function. However, soliciting personal preference will remove possible biases in these universal modelings.

---

> ### Author Response · Authors · 2022-11-18
> **pRMSE added to the main evaluation table**
>
> As per the recommendations, we have performed the average pRMSE evaluation across datasets and included them in the main evaluation Table as a part of the performance metrics. Additional violin plots with the distribution of pRMSE are also included in the appendix. We thank the reviewers for this suggestion.

---

### Decision · Program_Chairs · 2023-01-20

**Decision:**

Reject

**Justification For Why Not Higher Score:**

Number of issues regarding the experimental setting, assumptions and usability of the proposed methodology.

**Justification For Why Not Lower Score:**

N/A

**Metareview: Summary, Strengths And Weaknesses:**

This paper proposes incorporating user preferences as constraints into the recourse generation process. All the reviewers found the idea interesting and recognized the paper has merit. However, they all highlighted a number of issues regarding the experimental evaluation, practical use of the methodology and some of the assumptions. The response/discussion period did not clear out these concerns and I am unable to recommend acceptance.